# Coupling Molecular and Cellular Dynamics in a Large-Scale Monte Carlo Simulation

**DOI:** 10.3390/ijms262110763

**Published:** 2025-11-05

**Authors:** Jonah Chaiken, Amit Ifrach, Julia Sajman, Eilon Sherman

**Affiliations:** 1Racah Institute of Physics, The Hebrew University, Jerusalem 91904, Israel; 2Walder Department of Bioinformatics, Tal Campus for Women, Jerusalem College of Technology, Havaad Haleumi 21, Givat Mordechai, Jerusalem 91160, Israel

**Keywords:** MCell, Monte-Carlo simulation, whole cell modelling, immune synapse, T cells, microscopy, blender

## Abstract

Cells change their shape to survive, proliferate, and function. Such changes are both driven by stochastic molecular interactions and affect them in return. Recent Monte-Carlo simulations, such as MCell4, can explicitly capture the interactions of millions of molecules, yet cannot dynamically couple these interactions with changes in morphology. Here, we extend the MCell4 simulation platform by incorporating physical forces that allow bidirectional feedback between dynamic molecular interactions and outer or intracellular membranes. We start with some simple examples such as a moving piston and a fluctuating membrane. We then simulate the spreading of T cells on antigen-presenting cells or an activating surface due to cognate interactions of surface molecules, such as receptors and their ligands or integrins. The coupled simulation quantitatively accounts for the expected correlation of molecular interactions and the spreading dynamics of the cell surface. Thus, our approach provides a versatile foundation for simulating a variety of dynamic cell systems and processes.

## 1. Introduction

The dynamics of cellular structure emerge from intricate molecular interactions that drive fundamental biological processes essential for life. Coordinated networks of protein–protein interactions, membrane dynamics, and signaling cascades enable cells to respond, adapt, and function within their environment [1]. These interactions underlie the structural plasticity required for processes such as cell division, locomotion, secretion and uptake, protein recycling, membrane fusion, cytoskeletal reorganization, and organelle trafficking.

A particularly striking example of a highly dynamic cell structure is the immune synapse (IS) formed between T cells and antigen-presenting cells (APCs) [2].

As central orchestrators of adaptive immunity, T cells must rapidly decide whether to mount responses against foreign antigens while maintaining tolerance to self. These decisions require extraordinary specificity and sensitivity, leading to outcomes such as proliferation, differentiation, cytokine secretion, or cytotoxicity. Dysregulated signaling can result in autoimmunity or graft rejection, whereas insufficient signaling may cause anergy. How T cells integrate multiple molecular cues at the IS into accurate and reliable immune decisions remains a central question in immunology.

The IS exemplifies how molecular interactions generate dynamic cellular organization. Engagement of T cell receptors (TCRs) with peptide–MHC complexes initiates the formation of supramolecular activation clusters (SMACs), with TCR–CD3 complexes concentrating centrally (cSMAC) and integrins such as LFA-1 forming a surrounding peripheral ring (pSMAC) [3,4]. This “bull’s-eye” pattern, supported by actin cytoskeletal remodeling [5], stabilizes the T cell–APC interface. Synapse formation proceeds rapidly, reaching maximal organization within ~30 min, and is subject to temporal regulation that shapes the ensuing immune response.

Within the IS, molecular interactions are tightly coupled to the physical topography and dynamics of the interface. Receptor–ligand binding and adhesion molecules such as integrins [6,7] drive assembly, while surface architecture reciprocally regulates molecular distribution: protrusions enrich certain molecules [8], whereas bulky proteins may be excluded from close contacts [9]. Reaction–diffusion processes and motor-driven transport further refine the spatiotemporal organization of proteins, giving rise to the rich and dynamic patterning observed at the IS [3].

Since the underlying mechanisms of IS formation and T cell activation are under intensive study, simulations have been developed to complement experiments and shed light on the mechanisms that affect molecular patterning and IS formation [10,11,12]. More recently, we have shown that detailed microscopy images of the immune synapse can provide starting conditions and constraints for quantitatively predictive simulations of molecular organization at the IS [13], whereas previous computational modeling and simulations could capture only relatively small segments of the IS (~1 µm^2^), with a limited number of molecular species (<10) and copy numbers (typically, <100K). Monte-Carlo-driven simulations of significantly larger scale, such as MCell3 [14] and MCell4 [15], have been developed and can potentially serve for broader, and more detailed descriptions of the IS. However, to date, these simulations have not been able to dynamically modify the simulated surfaces in response to molecular interactions. Thus, such simulations cannot readily capture the complex dynamics of signaling and cell reorganization that occur at the immune synapse and enable T cell decision-making.

Previously, we introduced physical forces into MCell4 [15] to enable large-scale simulation of the IS. Here, we provide a demonstration of the feedback between the binding of freely diffusing receptors and ligands on the surface of interacting cells with their target cells or surfaces. Molecular interactions reinforce the tight interface, while bringing the interacting surfaces close to each other facilitates additional receptor–ligand interactions. We expect that the various types of forces that we have introduced into the large-scale MCell4 simulations will become a valuable tool for enhanced simulations, and thus better understanding of immune synapses as well as additional dynamic cellular systems. Furthermore, this tool will provide a framework to add physical forces for an ever-increasing complexity of cellular simulations of various cell types, such as neurons, myocytes, and epidermal cells.

## 2. Results

### 2.1. Our Approach for Large-Scale Monte-Carlo Simulations with Added Forces

MCell4 is a relatively new implementation to MCell in C++ that provides a Python API [15]. It also enables custom biochemical reaction modeling through native support for BioNetGen (BNGL) [16] species, reactions, and the rate constants associated with the given reactions. Further, it allows parallelization and easier extensibility. Using MCell4 in conjunction with the Blender add-on ‘CellBlender’ for visualization, one can produce highly detailed simulations with elaborated meshes containing millions of nodes and edges. Moreover, highly realistic meshes can be generated with AI support, purchased online, or captured by microscopy, to be used in simulations.

Together with the MCell4 and CellBlender basis, we aimed to integrate multiple force types, including elastic forces of cell membrane stretching, the kinetic forces that are exerted by molecules impinging on segments of the membrane, thermal fluctuations, and inner or external pressure. The goal of this integration was to simulate dynamics of complex cell processes, focusing on T cell spreading on an activating surface as an outstanding example. The required forces for such a simulation were first created and tested separately through simple configurations. Each force adds to the complexity of object dynamics that can be encapsulated in a single simulation.

### 2.2. Integration of Multiple Force Types into MCell4

The first force introduced was kinetic molecular interactions between a planar rigid membrane with freely diffusing and non-interacting particles, such as atoms of an ideal gas (Figure 1A). This can be viewed as a simplified model of a piston relying on particle collisions. The particles were free to move in the top and bottom compartments of the piston chamber (Figure 1A). All particles were given a mass. Next, various numbers of particles were released in each compartment, ranging from 100 to 100,000. The number of released particles could be essentially much larger, yet runtimes increase monotonically with this number. For simplicity, the piston membrane in the middle of the chamber was not given a mass. The volume of the piston chamber was of a cube with side length of 0.5 μm.

In theory, the piston membrane can be assumed to move in the manner of a one-dimensional random walk when both the top and bottom particles are of equal mass and count. Using this knowledge, we recorded the resulting z-axis movement of the piston membrane for various masses and particle counts.

First, bottom molecules were given a 10-fold higher mass than the top molecules. As can be seen, the desired general outcome was obtained of moving the piston membrane up via drift (Figure 1B). The motion of the piston membrane was reversed when the mass ratio of the top and bottom molecules was reversed—namely, the top molecules were given a 10-fold higher mass than the bottom molecules (Figure 1C).

Next, we released an equal number of particles with equal mass in each compartment. This configuration resulted in the expected one-dimensional random walk of the piston membrane (Figure 1D). There appeared to be a random and miniscule amount of movement on the order of nanometers caused by the kinetic forces of the impinging particles. The calculated diffusion coefficient of this motion was 6.53 × 10^−2^ nm^2^/μs. For each simulation, we also calculated the relative work done by the system (*W*), given the changes in volume of the top and bottom compartments of the piston chamber under isothermal conditions: $W=nRTln(\frac{V_{2}}{V_{1}})$. *V*_2_ and *V*_1_ are the time-dependent and initial volumes of either one of these compartments, respectively.

Having incorporated the interactions of particles with a mesh to MCell, we next added elastic stretching forces to the simulation. Such stretching forces also enabled the membrane to fluctuate, given some ambient temperature. The energy calculated was due to the extent a face was stretched relative to a predefined relaxed position. Movement was again subject to the Metropolis criterion for changes in the position of membrane nodes.

The elastic forces were tested by stretching a single point on a flat flexible membrane, which underwent thermal fluctuations (Figure 2A, left image). Upon release of the stretching force, the spike height dropped (Figure 2A, middle and right images). The peak membrane height dropped over time in nearly a linear fashion (Figure 2B), likely due to thermal dampening of its motion

### 2.3. Simulations of Interacting Cells

We next used MCell4 with its integration of forces to model the dynamic interface between cells, such as the immune synapse that forms between a T cell and an APC. Previously, we have captured such interactions using fluorescence confocal microscopy (Figure 2C) [17]. In this example, the T cell and the antigen-presenting cell are each labeled with a different stain, adhere to opposing surfaces, and are brought into contact under the microscope [17] (see Methods). This imaging approach allows for high-resolution imaging of the cells’ interaction with its resultant interface—i.e., the immune synapse.

In a fairly simple realization, each cell was first modeled as an approximated sphere (i.e., icosphere), with membrane ruffles. The cells’ model included surface molecules that diffuse and interact. There were 10^4^ and 10^5^ surface molecules, mimicking T cell antigen receptor (TCR) molecules on the surface of one sphere and peptide-MHC (pMHC) molecules on the surface of the second sphere. In the simulation, the two spheres were brought into close contact as an initial condition. Upon encounter, one sphere was able to plastically deform the surface of the other sphere, forming a tight and dynamic contact.

To approach a more physiological appearance, we applied a uniform force (through an increase in internal pressure) that drove one cell to contact the other more closely, showing induced spreading, aside from merely plastic deformation. The geometric configuration then came to a steady state to allow for receptor–ligand interactions to dominate the cell-to-cell interaction.

### 2.4. T Cell Spreading: Towards Realistic Simulation of the Immune Synapse

Next, we aimed to simulate an even more realistic configuration of an experiment involving a T cell spreading on an activating coverslip [18]. We follow a similar imaging approach to the previous example, only now letting the cell interact with the coverslip and its resultant spreading. This experiment involves the adherence of the T cell to an upper surface. Next, the surface and cells are placed upside down onto a glass bottom surface coated with an activating antibody (typically, anti-CD3e). Here, we aimed to capture the initial contacts between the T cell and the activating surface. Thus, the upper surface was separated from the bottom coverslip with beads with a 20 µm diameter (47148-10, Corpuscular), which is larger than the cells’ size (typically, 12–15 µm). The cells were fixed, labeled using αCD45 Alexa647 (304056, BioLegend, San Diego, CA, USA), and fluorescently imaged using an Abberior confocal microscope equipped with a high (×100)-magnification objective (see Section 4). The cells show enrichment of membrane structures at the lower side of the cell (Figure 3A, maximal intensity projection). For incorporation into MCell4, we converted the cell image into a 3D mesh. Faces with a typical length smaller than 1 µm were smoothed out. This rendering shows pronounced membrane protrusions, including microvilli and lamellipodia, directed toward the bottom surface (Figure 3B; side and bottom views).

Next, we established a physical model of cell spreading, following Wahl et al. [19]. In this model, polymerized actin pushes normally on the face of the plasma membrane. The polymerized actin undergoes inward motion due to myosin forces; thus, the actin mesh undergoes a treadmilling motion. The membrane tension balances the actin mesh force on the membrane, resulting in its fluctuating motion. Only upon contact with the activating surface, surface molecules on the cell membrane (namely receptors and integrins) can specifically bind their molecular partners (cognate pMHCs and integrins, respectively) embedded on the surface of the coverslip. These interactions create effective friction forces that lead to extension of the cell contacts with that surface and its eventual spreading.

In our simulation, we included up to 250K surface molecules on the cell surface, and an identical number of binding molecules on the activating surface. For simplicity, we simulated the actin forces implicitly, yet membrane tension and molecular interactions were simulated explicitly. We then let the simulation run for 1000 iterations, allowing the cell to be brought into contact with the surface, while its surface molecules could diffuse and interact with that surface. During the simulation, we routinely captured the location of the surface node coordinates and the diffusing molecules and their interaction with the surface molecules.

We show the time evolution of the cell spreading on the coverslip (Figure 3C; Movie M1). As expected, the number of contacts and the spreading area grew monotonically over time (Figure 3D,E). The relationship between the growth of these parameters was non-trivial and showed an earlier rise in molecular interactions (Figure 3E, orange curve), but faster growth in contact area (blue curve). This curve of area growth could be further compared with experimental measurements taken using single molecule localization microscopy of Jurkat cells engaging a TCR-activating coverslip [13]. For that, the simulation dynamics and time were scaled to correspond to the experimental time. Strikingly, our simplified simulation could roughly capture the dynamics of the measured area growth (Figure 3F).

Taken together, we demonstrate how the iterative coupling of molecular interactions with the cell membrane can lead to the emergent process of T cell spreading on functional surfaces, all within a realistic configuration defined by microscopy.

## 3. Discussion

Over the past two decades, high- and super-resolution microscopy have been used to image TCR-dependent signaling complexes in single molecule detail in fixed and live T cells [20,21,22]. Thus, we and others have recognized multiple mechanisms that are critical for robust TCR triggering [23]. Such mechanisms include receptor clustering [21,24], conformational changes in receptor chains [25,26,27], dynamic formation of signaling complexes [20], cooperativity in triggering within clusters [28,29], physical segregation of glycoprotein-phosphatases [9,13,30], and effects of cell topography [30,31]. A major effort in the field is to integrate such mechanisms, thus accounting for the unique properties of T cell recognition—i.e., their sensitivity, selectivity, and speed [32]. Still, current computational modeling and simulations cannot capture the complex dynamics of signaling and cell reorganization that occur at the immune synapse and enable T cell decision-making. Similarly, other complex and dynamic cellular processes that involve molecular interactions cannot be readily simulated and require state-of-the-art modeling approaches and computational power (e.g., [33,34,35,36]).

Here, we integrated physical forces into the Monte-Carlo simulation MCell4. This simulation allows modeling of finite numbers of molecules within realistic cellular geometries and offers some advanced features: integration with Blender for arbitrary mesh generation, support for cytosolic and membrane-bound proteins, stochastic state transitions, multiple mobility modes (diffusion, drift), and batch simulations for parameter scanning. A known limitation is its inability to simulate dynamically changing meshes, which we address in this work. Our forces include the interaction of molecules with membrane meshes as well as membrane stretching, thermal fluctuations, and pressure. Such forces enable elaborate dynamic simulations that encapsulate the chemistry and physics of cell-to-cell interactions. In our case, we simulated such interactions leading to the physiological outcome of T cell spreading on an activating surface. For that, we had to incorporate the variety of these added forces into MCell4. Arguably, our model is a small and limited biological encapsulation of cell signaling demonstrated in MCell4. Moreover, full-scale simulation of the immune synapse is clearly more complex, as it involves interactions of many more molecules.

Other than Monte-Carlo simulations such as MCell4 and Smoldyn [37], numerous simulation tools have been developed to study molecular interactions within cells, including signaling pathways and enzymatic reactions. Traditional approaches often rely on ordinary differential equations (ODEs) or cellular automata, assuming complete molecular mixing or compartmental averaging—both of which can obscure critical spatial variations in protein concentrations. Tools like Virtual Cell (VCell) [38] integrate spatial constraints from 2D and 3D microscopy and solve both ODEs and partial differential equations (PDEs). However, they fall short in capturing molecular heterogeneity and the inherent stochasticity of diffusion and interactions at the single-molecule level. On the other hand, Network-Free Stochastic Simulator (NFsim) [39] is a powerful tool for simulating large and combinatorially complex biochemical systems, especially where traditional network-based methods (full reaction network generation) become infeasible due to the explosive number of possible molecular species and states. Still, it can capture only well-mixed realizations and cannot support geometry-based models. Alternative approaches to cell simulations may also include classical molecular dynamics simulators such as Large-Scale Atomic/Molecular Massively Parallel Simulator (LAMMPS) [40]. These simulations are highly scalable and parallelized and support molecular dynamics, coarse-grained models, and custom force fields. They can readily simulate mechanical, thermal, and electrostatic interactions; however, they are not inherently designed for stochastic reaction–diffusion, which requires more effort to model biochemical reactions as well as realistic biological geometries.

Future effort could integrate additional molecular species on the surface (co-receptors, glycoproteins, etc.), subcellular entities (e.g., nucleus, cytoskeleton, vesicles, actin filaments), and intracellular molecules (e.g., for signaling). Additional future forces may include object twisting and bending, as its energy cost becomes dominant when modifying membrane surfaces with high curvatures (R^−1^ > 1/50 nm^−1^) [41,42].

Our simulations were conducted primarily on a desktop PC. There exists a computational limit without using a significantly large cluster or a supercomputer. On a PC computer containing an Intel i7 processor, Nividia Geforce GTX 1650, and 32 GBs of RAM, we get runtimes ranging from minutes to days, depending on the number of interacting molecules (Table 1) and nodes, with their associated algorithms and interactions.

Simulations can also be done with clusters or supercomputers with much more computing power than any computer equivalent to the one discussed above. For instance, the Abacus Summit cosmological N-body simulation suite simulated a total of 6 × 10^13^ particles [43] using a supercomputer. There are approximately 2 × 10^10^ water molecules in an *E. coli* bacterium and 10^9^ lipid molecules in the cell membrane of a small eukaryotic cell [1], adding to a total of approximately 2.1 × 10^10^ particles that must be simulated in this mixed cell. This puts simulation of an entire eukaryotic cell close to the limit of some of the best supercomputers today.

Our integrated simulation framework can simulate multiple other intercellular and intracellular interactions, beyond the shown examples. Such examples may include vesicles transporting material within and across the cell, cellular signaling networks, and passive and active transport in various cell types. This is in part due to Blender allowing the user to create realistic models of high triangulation density. We conclude that our approach provides a versatile foundation for simulating a variety of dynamic cell systems and processes.

## 4. Methods

The Python interface used allows for a high degree of flexibility. Full control of the geometry is allowed throughout the simulation (Appendix A). Definition of various reactions uses the new BioNetGen library added to MCell4. Parallelization is defined by determining the 3D partition size in Python. This allows for further optimization of runtimes in the simulation.

Installation of the simulation, instructions, and examples can be found in the following link: jchaiken12/3D_cell_simulation: GitHub repository for research paper “Towards large-scale 3D modelling and simulation of the immune synapse” (https://github.com/jchaiken12/3D_cell_simulation; URL accessed on 30 October 2025)

### 4.1. Detailed Molecular Simulation

**Requirements**. For ease of use, employing a standard PC with an i7 processor, a GPU, and at least 16GBs of RAM is advisable. The physics is coded in Python version 3.9. The users should follow the directions on MCell’s website on how to download MCell4 given their operating system. The structure of the simulation is depicted in Figure 4 and is explained in detail below.

**Simulation Setup**. Each simulation requires multiple input files (Figure 4). The geometry.py defines the Euclidean positions of the nodes along with their edges. This must be a triangulated mesh. The physical mesh is the triangulated mesh created in Blender to be exported with a custom export function to create the geometry file. The stereometry.py file contains some useful geometric functions used to create the simulation. The membrane_physics.py contains energy and elastic displacement calculations used in the model.py file. The parameters.py file defines the number of iterations, pairing distance, time step, random seed of the simulation to be run, and any other needed module-level variable of the simulation. The model.bngl file defines the reactions and their associated reaction rates. The bngl_molecule_types_info.py file defines if the molecules are 3D or 2D surface molecules and their given diffusion speeds. Next, the instantiation.py file defines the initial release location and adds the molecules and geometries to the simulation. The observables.py handles the export of the visualization data. The subsystem.py file loads in the model.bngl file with its reaction and rate along with additional information such as diffusion constants for loaded elementary molecule types.

First, the simulation is run by calling the main execution file named model.py in the terminal. This file uses all the files above to be woven together to produce a physical simulation of a cell of the user’s choice. A k-d tree is used to determine distances of static molecules distributed on the coverslip to molecules diffusing on the cell. Once the simulation is finished, the user references the newly created visualization data and runs Blender with the CellBlender add-on again in the terminal. This produces the desired simulation to be visualized in Blender. This process is repeated for any simulation that the user creates.

**Input**. Input parameters include parameters that describe the physical properties of the interacting cells and of the molecules that interact within and across the interfaces. The cell meshes are made by the user on Blender and extracted to create a custom cell-to-cell interaction. Parameters of our cell-related simulations are listed in Appendix A.

**Simulation Main Run File**. As stated above, the model.py file is the main run file. This contains the energetics of the simulation explained in more detail in a later section. The surface molecule interactions are defined within MCell4. Briefly, in our simulation we assume specific Hamiltonians of a quasi-equilibrium system and with mean-field approximations. The simulation simplifies noncrucial elements in the cell-to-cell interaction considering only their general effects. Such entities include lipids on the membranes and water molecules, which are not specifically described in the simulation. In contrast, protein molecules of interest are simulated. The simulation algorithms are done using Monte-Carlo simulations using Metropolis criterion to determine the probability between possible configurations.

**Outputs**. The positions and energies of each region of the cell are calculated, maintained, and updated in each iteration. The surface proteins’ positions are also recorded in each iteration. Visualization output is made available using CellBlender after the main run file finishes running by a command on the terminal, referencing the newly created data. This ultimately produces a movie in CellBlender.


**Monte Carlo Simulations**


**Simulation Energetics**. In the simulations we used the Hamiltonian *H = H_int_ + H_el_* to calculate the energetics of the overall interactions between the T cell membrane and the APC membrane or the activating surface. A bending component, *H_bend,_* could be added in the future, as detailed below. The interaction part, *H_int_*, is defined as follows:(1)$$H_{int}=\sum_{i}^{n}\frac{1}{2}m_{i}v_{i}^{2}$$
where each particle is represented as *i*. The movements were made by changes in the momentum, *P_int_*, incurred by each particle collision with the membrane. The momentum is defined as follows:(2)$$P_{int}=\sum_{i}^{n}m_{i}v_{i}$$

The elastic part of the Hamiltonian, *H_el_*, is defined as follows [41]:(3)$$H_{el}=\sum_{i}^{n}\frac{\kappa }{2a_{i}^{2}}{\left(∆a_{i}\right)}^{2}$$
where $\kappa =\kappa_{1}\cdot \kappa_{2}/\left(\kappa_{1}+\kappa_{2}\right)$ is the general effective bending rigidity of two membranes. In this case, the bending rigidity is effectively $\kappa \approx \kappa_{1}$, since $\kappa_{2}\gg \kappa_{1}$ and is simulated at different values. The area, $a_{i}$, is defined for each face. The new area is determined by attempting another move and calculating the new area after the potential move is made.

In the current implementation of the forces, we have not incorporated bending energies. In the future, the bending part of the Hamiltonian, *H_bend_*, could be defined as follows [41]:(4)$$H_{bend}=2{\kappa_{b}M}_{2}$$

This is the Helfrich bending energy [44], assuming our model is minimal, i.e., not a bilayer couple model, area-difference elasticity model, or a spontaneous curvature model. This assumption is valid given our membrane in the simulation does not have two layers. $\kappa_{b}$ is the bending elastic constant.

The moment M_2_ is defined as follows:(5)$$H_{el}=\sum_{i}^{N_{\nu }}{\frac{1}{4}a_{i}\left[{\left(\nabla_{s}^{2}x\right)}_{i}\right]}^{2}$$
where *a_i_* is the area of triangles around node *i*, *N_v_* is the total number of nodes, *x* is the surface coordinate, and $\nabla_{s}^{2}$ is the discrete Laplace–Beltrami operator. Lastly, the membrane would move proportionally to the energy related to the pressure. The displacement was determined by setting an initial volume of the cell and assuming pressure remained constant. The displacement due to pressure is defined as follows:(6)$$D_{P}=c\frac{{\left(∆V\right)}^{2}}{V_{int}}$$
where c is the displacement constant.

**Simulation Dynamics**. The simulation propagates in time by iterations of 1 μs. In every iteration all molecules and nodes of the mesh attempt to move. The nodes of the mesh have the following rules for movement:The node is not outside the movement space and not crossing another mesh.The probability of the move according to the Metropolis criterion is as follows:(7)$$P(old\;state\to new\;state)=\{\begin{array}{cc} 1 & \Delta E<0\\ exp(-\Delta E) & \Delta E>0 \end{array}$$

### 4.2. Sample Preparation and Confocal Microscopy

For imaging cell–cell conjugates (Figure 3B), we employed our previously published approach [17]. Briefly, we let Jurkat J76 (CD8+) and T2 hybridoma cells (both a kind gift from the Acuto lab at Oxford) interact. Twenty-four hours prior to imaging, the T2 cells were loaded with NY-ESO-1 peptides, thus serving as APCs. Each cell type adhered to a different surface: the bottom of an 8-well ibidi #1.5 coverslip (Gräfelfing, Germany) and a small glass that fitted into the well. Adherence to the glass surfaces was promoted by coating them with non-stimulatory antibodies (αCD45 and αCD11a; PMG555480 and 555378, respectively, by BD Pharmingen, San Jose, CA). The cells were brought into contact as the small glass was placed (upside down) on the bottom glass of the well. The upper surface was separated from the bottom coverslip with beads with a 20 µm diameter (47148-10, Corpuscular, Cold Spring, NY, USA), which is larger than the cells’ size (typically, 12–15 µm). For imaging, the plasma membrane of the cells was stained for αCD45 and Alexa647 (304056, BioLegend, San Diego, CA, USA) and the PM of the T2 cells was stained using DPEE-Atto565. Prior to imaging, the cells were fixed by 2.4% paraformaldehyde (PFA) for 30 min in 37 °C and washed with PBS.

For imaging cell protrusions (Figure 3C,D), the cells were attached to an upper coverslip, which was placed on the bottom of a well without any cells. The same beads were used for spacing the two surfaces. The upper glass was placed into an ibidi well. The cells were fixed and labeled with an αCD45 antibody (as above).

Single and two-color fluorescent confocal microscopy was carried out using an Abberior STED/confocal microscope (Expert line; Abberior Instruments, Göttingen, Germany), mounted on a TiE Nikon microscope (Nikon Instruments, Amsterdam, The Netherlands) and operated by the Imspector software (v0.13.11885; Abberior Instruments, Göttingen, Germany). The microscope was equipped with a high-magnification (×100, 1.49 NA) oil immersion objective (CFI SR HP Apochromat TIRF, Nikon Instruments, Amsterdam, The Netherlands). Samples were excited with either a 2 mW 561 nm pulsed laser (60 ps) or with a 2 mW 640 nm pulsed laser (60 ps) at 10%. The pinhole was set to 1 Airy unit. Three-dimensional imaging used a piezo stage to scan the area with 1 μm axial resolution.

## Figures and Tables

**Figure 1 ijms-26-10763-f001:**
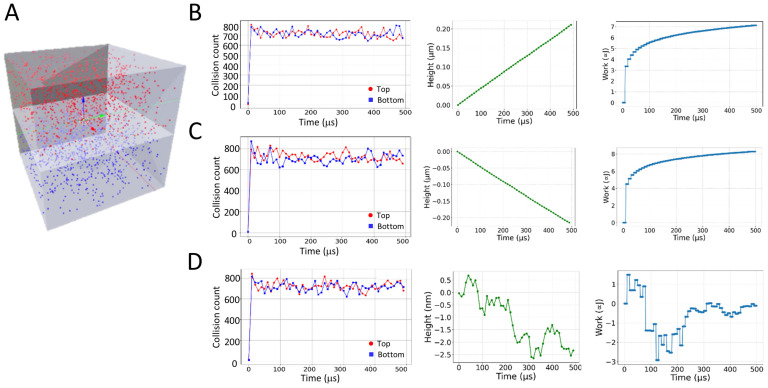
Motion of a piston membrane due to particle interactions. (**A**) Illustration of the piston model. The model was constructed in Blender 2.93 using MCell4. (**B**–**D**) Shown are the collision counts of top and bottom molecules (in red and blue lines, respectively), height of the membrane, and work performed by the system over simulation time for various ratios of particle masses. (**B**) Simulation with top molecules with masses of 0.01 undefined mass units and molecules at the bottom of 0.1 undefined mass units. (**C**) Simulation with top molecules with masses of 0.1 undefined mass units and molecules at the bottom of 0.01 undefined mass units. (**D**) Simulation with all molecules of masses of 0.1 undefined mass units.

**Figure 2 ijms-26-10763-f002:**
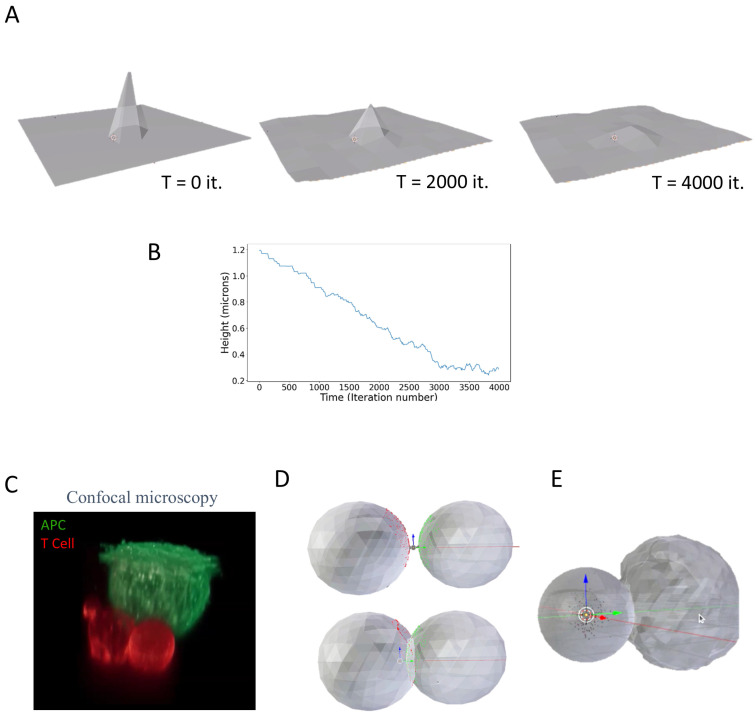
Modeling membrane stretching and cell–cell interactions. (**A**) Illustration of a membrane model. The model was constructed in Blender 2.93 using MCell4. (**B**) The peak height of the membrane over time. (**C**) A confocal fluorescence image of a T cell (red) engaging an antigen-presenting cell (green). The T cell’s diameter is ~12 μm. (**D**) A simplistic model of cell–cell conjugates using interacting icospheres, with one of the icospheres undergoing plastic deformation. (**E**) A simplistic model of cell–cell conjugates using interacting icospheres, with one of the icospheres undergoing deformation and induced spreading due internal pressure forces.

**Figure 3 ijms-26-10763-f003:**
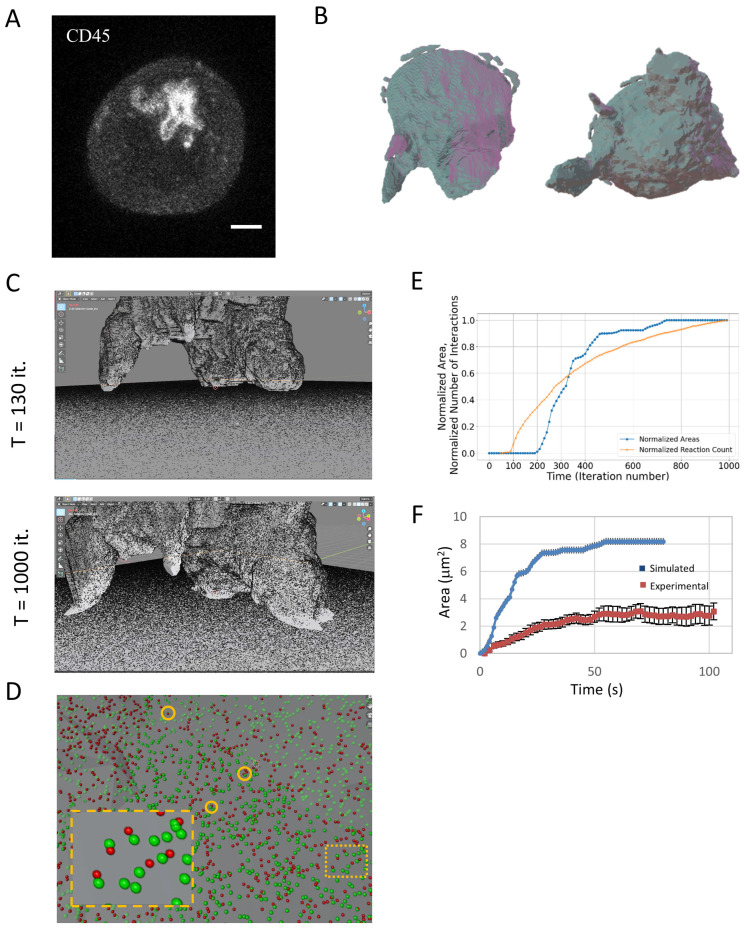
T cell spreading. (**A**) Fluorescence confocal imaging of a T cell engaging a coverslip coated with a TCR-activating antibody (αCD3ε). Scale bar—2 μm. (**B**) Three-dimensional rendering of the cell. Shown are side and bottom views (on left and right, respectively). (**C**) Snapshots from the beginning and the end of a movie of the simulation of the T cell as it engages the coverslip. 250K molecules were embedded on each of the surfaces. Molecules on the surface of the cell were allowed to diffuse and interact with the molecules on the flat surface. (**D**) A zoom onto the interface of the T cell and the coverslip, showing examples of molecular binding events (orange circles). (**E**) The time-dependent evolution of the normalized contact area and normalized cumulative number of molecular interactions. (**F**) A comparison of the simulated area growth (**E**) and experimental measurements of contact area growth using super-resolution microscopy, adapted from Ref. [13] (Figure 3J). The simulation dynamics and time were scaled to correspond to the experimental time.

**Figure 4 ijms-26-10763-f004:**
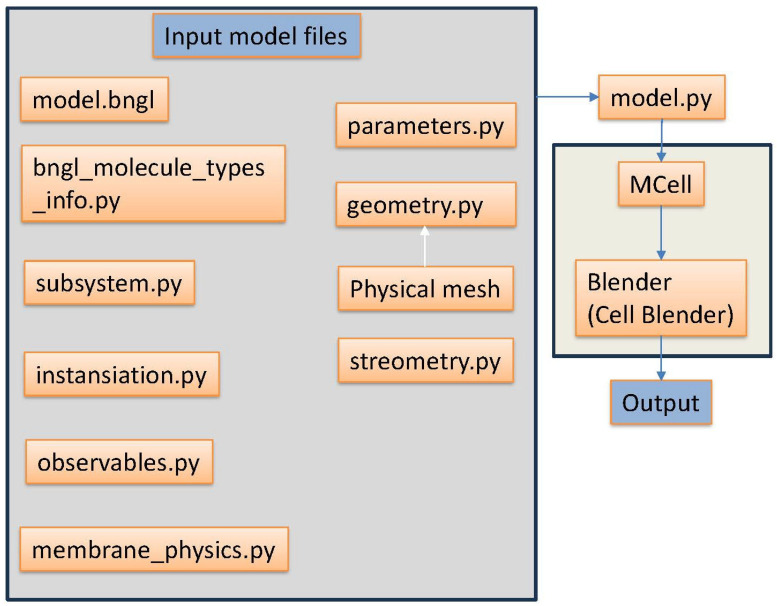
A schematic overview of the simulation. Files and outputs are produced sequentially in the order of the arrows.

**Table 1 ijms-26-10763-t001:** Simulation speed.

Total Number of Molecules *	Set Time	Simulation Speed [Iterations/Sec] **
4K	126.6 s	0.9
100K	2.35 h	0.161
500K	65 h	0.038

* Molecules are spread evenly on the surface of the two cells; ** number of nodes was ~12,000 for all simulations.

## Data Availability

The authors declare that the data supporting the findings of this study are available within the article and its Appendix A or are available upon reasonable requests to the authors. All custom source code is available from the specified GitHub repository.

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
