# Peer review of "Coupling Molecular and Cellular Dynamics in a Large-Scale Monte Carlo Simulation"

_ijms, 2025, doi:10.3390/ijms262110763_

Round 1

Reviewer 1 Report

Comments and Suggestions for Authors

This study presents a valuable extension of the MCell4 Monte-Carlo simulation platform by integrating physical forces to enable bidirectional feedback between molecular interactions and membrane dynamics. The simulation of T cell spreading and immune synapse formation were carried out and discussed in details. The work is valuable and meaningful. Revision suggestions are as follows:

  1. 1. In Section 2.3, when describing the cell-cell conjugate model, can authors provide quantitative data (e.g., contact area change over time, binding rate of TCR-pMHC) instead of only qualitative descriptions to strengthen the evidence for interface dynamics.
  2. In Section 2.4,it willenhance results reproducibility if the specific values of key parameters (e.g., diffusion coefficient of surface molecules, membrane tension coefficient) used in the T cell spreading simulation can be provided.
  3. In Section 4.1, the authors note that bending energy Hbend was not incorporated in the current model but do not explain why.What is the rationale for omittingHbend (e.g., computational cost, lack of parameter data)? How this omission might bias results (e.g., underestimating membrane rigidity).
  4. The authors claim the modified MCell4 framework advances large-scale simulation but do not compare it to other state-of-the-art platforms (e.g., LAMMPS, COMSOL, or Smoldyn). A comparative analysiswould highlight the work’s contributions and identify gaps for future improvement.
  5. 5.In the References section, correct the formatting of incomplete references (e.g., References 8, 9, 17 ... lack full volume, page) to comply with standard academic citation norms.

Reviewer 2 Report

Comments and Suggestions for Authors

The manuscript “Coupling Molecular and Cellular Dynamics in a Large-Scale Monte Carlo Simulation” by Chaikin et al presents an interesting idea of how to incorporate forces and cellular mechanics into simulations that typically focus on biochemical reaction modeling. As currently presented, I recommend against publication. That being said, it seems probable that most of these issues could be addressed. Below I list my concerns in roughly the order they appear.

  • In the quote “… could capture only relatively small segments of the IS (~1m^2)” do the authors mean square meters or is the unit mislabeled?
  • Figure 1 seems to be mislabeled. There is no C or D and B seems to be in the wrong spot.
  • It is not clear what the energetics in Fig. 1 are supposed to convey and they are not discussed in the manuscript.
  • In the caption of Fig. 1, D is listed as having mass of some particles be 2x10^10 units. Is this accurate? That seems more like the sanity check discussed in the manuscript than the data presented.
  • The predicted MSD is the same for balanced and unbalanced masses is the same. Is this expected? Where does the predicted MSD come from?
  • How is the calculated MSD related to the simulations shown? Heights either move on the order of 10^-9 units (meters??) or on the order of 0.1 (again, meters??).
  • What is the “uniform force (through increased internal pressure)” that the authors apply in the discussion of figure 2? Is it just ad hoc? Is it triggered? How is this induced spreading? Or is it just showing you could implement such a feature to show induced spreading?
  • The units are all mm when discussing the microscopy experiments and I assume they should be um.
  • When I tried to look up the beads to just confirm the sizes, the company listed does not exist. This seems to be a copy paste error from a previous article. I assume the company they meant to credit is “Corpuscular”.
  • Some sort of scale bar on the microscopy images would be useful.
  • In Fig. 3e, can the authors provide some sort of experimental reference? Even if it has to be scaled, something would be useful. Yes, the spreading changes but does it in anyway resemble a measured spread area? The other data in the paper is largely phenomenological. Having at least some comparison to externally measured data would be useful to show the models as implemented might approximate a real system.

Round 2

Reviewer 2 Report

Comments and Suggestions for Authors

The authors have addressed my concerns. I did notice two typos in re-reading the manuscript:

Figure 1: The labels C and D are missing in the figure

Line 125: “Bottom molecules were given a (10-fold) higher mass than bottom molecules.” Bottom is repeated twice and I suspect the brackets could be removed.

Author Response

The authors have addressed my concerns.

Response: We greatly appreciate the support and comments of the reviewers.

I did notice two typos in re-reading the manuscript:

Figure 1: The labels C and D are missing in the figure

Response: We have now added the missing labels of panels C and D in Figure 1.

Line 125: “Bottom molecules were given a (10-fold) higher mass than bottom molecules.” Bottom is repeated twice and I suspect the brackets could be removed.

Response: The text in this line has now been corrected to: “First, bottom molecules were given a 10-fold higher mass than the top molecules.”